ScanFold 2.0: a rapid approach for identifying potential structured RNA targets in genomes and transcriptomes

http://orcid.org/0000-0003-0275-0019 Andrews Ryan J. 1
http://orcid.org/0000-0002-5158-7572 Rouse Warren B. 2
O’Leary Collin A. 2
Booher Nicholas J. 3
http://orcid.org/0000-0001-6419-5570 Moss Walter N. 2 wmoss@iastate.edu
1 Department of Biochemistry, University of Utah , Salt Lake City, UT , United States
2 The Roy J Carver Department of Biochemistry, Biophysics and Molecular Biology, Iowa State University , Ames, Iowa , United States
3 Infrastructure and Research IT Services, Iowa State University , Ames, IA , United States
Karakülah Gökhan
Electronic publication date: 2022 Nov 8
Publication date: 2022
Volume: 10
Electronic Location ID: e14361
Received 2022 Jul 27; Accepted 2022 Oct 18
Copyright: © 2022 Andrews et al.
Copyright year: 2022
Copyright holder: Andrews et al.
License: This is an open access article distributed under the terms of the Creative Commons Attribution License, which permits unrestricted use, distribution, reproduction and adaptation in any medium and for any purpose provided that it is properly attributed. For attribution, the original author(s), title, publication source (PeerJ) and either DOI or URL of the article must be cited.
License URL: https://creativecommons.org/licenses/by/4.0/

Keywords: RNA, Motif discovery, RNA structure, Genome annotation, Sequence analysis

Funding: National Institute of General Medical Sciences R01GM133810 National Cancer Institute F31CA257090 This research was supported by National Institute of General Medical Sciences R01GM133810 to Walter N. Moss and National Cancer Institute F31CA257090 to Warren B. Rouse. The funders had no role in study design, data collection and analysis, decision to publish, or preparation of the manuscript.

==============================
A major limiting factor in target discovery for both basic research and therapeutic intervention is the identification of structural and/or functional RNA elements in genomes and transcriptomes. This was the impetus for the original ScanFold algorithm, which provides maps of local RNA structural stability, evidence of sequence-ordered (potentially evolved) structure, and unique model structures comprised of recurring base pairs with the greatest structural bias. A key step in quantifying this propensity for ordered structure is the prediction of secondary structural stability for randomized sequences which, in the original implementation of ScanFold, is explicitly evaluated. This slow process has limited the rapid identification of ordered structures in large genomes/transcriptomes, which we seek to overcome in this current work introducing ScanFold 2.0. In this revised version of ScanFold, we no longer explicitly evaluate randomized sequence folding energy, but rather estimate it using a machine learning approach. For high randomization numbers, this can increase prediction speeds over 100-fold compared to ScanFold 1.0, allowing for the analysis of large sequences, as well as the use of additional folding algorithms that may be computationally expensive. In the testing of ScanFold 2.0, we re-evaluate the Zika, HIV, and SARS-CoV-2 genomes and compare both the consistency of results and the time of each run to ScanFold 1.0. We also re-evaluate the SARS-CoV-2 genome to assess the quality of ScanFold 2.0 predictions vs several biochemical structure probing datasets and compare the results to those of the original ScanFold program.

Introduction

Interest in RNA has, arguably, never been higher. RNA plays key regulatory roles in all organisms including human pathogens such as HIV, Zika, and SARS-CoV-2 (Cao et al., 2021; Li et al., 2018; Watts et al., 2009). Furthermore, since both the viral vector and the most efficacious preventative modality for COVID-19 both consist of RNA, interest in RNA as both a therapeutic agent and target is surging (Bhat, Karve & Anderson, 2021; Damase et al., 2021). Significantly, in both its biological function and potential for targeting, RNA secondary structure plays key and diverse roles (Andrzejewska, Zawadzka & Pachulska-Wieczorek, 2020; Disney, 2019; Hargrove, 2020; Meyer et al., 2020; Szabat et al., 2020; Wan et al., 2011). For example, in processes such as RNA splicing and posttranscriptional gene regulation, secondary structures can vary the distances between or accessibility of various regulatory elements in RNA (Andrzejewska, Zawadzka & Pachulska-Wieczorek, 2020; Jiang & Coller, 2012; Li et al., 2014), as well as provide specific platforms for recognition by regulatory molecules (e.g., proteins and noncoding RNAs (Law et al., 2006; Sanchez de Groot et al., 2019; Yang et al., 2020)). Secondary structures are also found within long noncoding RNAs (Andrzejewska, Zawadzka & Pachulska-Wieczorek, 2020; Chillon & Marcia, 2020; McCown et al., 2019; Somarowthu et al., 2015) and in the coding regions of mRNAs, where there is increasing awareness of their roles in modulating translation and protein folding (Andrzejewska, Zawadzka & Pachulska-Wieczorek, 2020; Faure et al., 2016; Faure et al., 2017; Mauger et al., 2019; Mustoe et al., 2018).

Unsurprisingly, there is great interest in gaining additional structure-function knowledge about RNA (particularly as related to human health) and in therapeutically modulating RNA biology via its secondary structure. Both tasks require the identification of robust structural models of RNA folding which, for large genomes/transcriptomes, is an immense challenge. Several rapid and robust algorithms for RNA secondary structure prediction are available (Lorenz et al., 2011; Reuter & Mathews, 2010; Zuker, 2003) as well as powerful methods for assessing the phylogenetic significance of structure (Manfredonia et al., 2020; Rivas, Clements & Eddy, 2017, 2020). Similarly, there have been great advances in approaches for high-throughput probing of RNA secondary structure (Mitchell, Assmann & Bevilacqua, 2019; Regulski & Breaker, 2008; Smola & Weeks, 2018; Strobel, Yu & Lucks, 2018; Tomezsko, Swaminathan & Rouskin, 2021). Despite this, a major challenge that hampers efforts to understand and target RNA secondary structure, is the determination of which fragments form extremely stable, and likely functional structures.

Early on, it was noted that functional RNA structures have a sequence-ordered stability bias. That is to say, the predicted folding free energy of functional RNA is lower than that of randomized sequences (Clote et al., 2005; Moss, 2018; Qu & Adelson, 2012). This bias is quantified via the thermodynamic z-score, which reports the difference between the predicted minimum free energy (MFE) of folding for a native/ordered RNA and the expected MFE based on the nucleotide content alone (i.e., the native sequence is shuffled and refolded multiple times to calculate the mean and standard deviation of expected MFEs). The native MFE is subtracted from the expected and the resulting value is divided by the standard deviation (Eq. (1)) indicating the number of standard deviations more or less stable the native secondary structure is vs that predicted by nucleotide content (i.e., negative values indicate significantly ordered stability) (Andrews, Baber & Moss, 2017; Clote et al., 2005). Mono- vs di-nucleotide shuffling can affect the calculation of z-scores (Forsdyke, 2007). Dinucleotide shuffling preserves the nearest-neighbor nucleotides (i.e., nucleotides that can stack in helices) that are used in MFE calculations (Gesell & Washietl, 2008). Mononucleotide shuffling on the other hand, abolishes this pattern, and can potentially overestimate the magnitude of the z-score (Gesell & Washietl, 2008); however, our original analysis of shuffling methods used in ScanFold (Andrews, Roche & Moss, 2018), found little difference in predicted ordered structure and, indeed that mono-nucleotide results were slightly better supported by available data.

ScanFold 2.0 (SF2) uses the same approaches as ScanFold 1.0 (SF1) without the need for explicit MFE calculations of randomized sequences to determine thermodynamic z-scores. To bypass the computationally expensive explicit z-score calculations, we have implemented a machine learning approach: Google’s publicly available TensorFlow algorithm (Abadi et al., 2016a, 2016b). TensorFlow was trained using 20 different sequence features including: sequence length, GC percentage, CG ratio, AU ratio, and the frequency of 16 different dinucleotide types. Using these features, both mono- and dinucleotide shuffling models were generated. SF2 uses these models to estimate the randomized MFEs and standard deviations needed to calculate thermodynamic z-scores for all windows. This new version of ScanFold still uses the same algorithm to highlight local structural features, ScanFold-Fold (Andrews, Baber & Moss, 2020; Andrews, Roche & Moss, 2018), which is now the rate limiting step of the program. This improvement has led to an increase in computational speeds of at least 10×, and in some cases increases of over 100× (File S1). This new tool is available for download on GitHub (https://github.com/moss-lab/ScanFold2.0) or for use through a webserver hosted at: https://mosslabtools.bb.iastate.edu/scanfold2.

Methods

TensorFlow training of z-score model

An overview of the training process can be seen in Fig. 1. A total of 836,377 representative sequences were generated to be used for training. Sequence lengths were between 60 and 200 nt (based on typical ScanFold window sizes (Andrews, Baber & Moss, 2020; Andrews, Roche & Moss, 2018)) in 20 nt increments. To represent as many potential sequence types as possible, dinucleotide frequencies for all 16 dinucleotide types were set to vary between 0% and 45%, averaging ~6.3% across all sequences. Native MFEs, mean of 100 randomized MFEs ( MFE¯), and their standard deviations ( σ) were calculated for all sequences using RNAfold version 2.4.18 (Lorenz et al., 2011). Two different randomization procedures were used to train the algorithm: mononucleotide and dinucleotide shuffling (Andrews, Baber & Moss, 2020; Andrews, Roche & Moss, 2018; Gesell & Washietl, 2008). Twenty different training features were also collected for each sequence including: sequence length, GC percent, AU ratio, GC ratio, and all 16 dinucleotide frequencies.

Figure 1 Schematic of ScanFold 2.0 training procedure.

Representative sequences were generated for a range of lengths (between 60 and 200 nt) and dinucleotide frequencies. These sequences were shuffled and analyzed using RNAfold to determine their MFEs, mean MFEs and respective standard deviations. Mean MFEs and standard deviations were then combined with 18 sequence composition features to comprise all 20 training features. These 20 features were used to generate mean MFE and standard deviation models.

All 20 features were used during training of MFE¯and standard deviation ( σ) models. The mean MFE and STD models are Keras sequential, with one preprocessing normalization layer, and two hidden layers: Rectified Linear Unit (ReLu) and sigmoid. RNAfold is used to calculate MFEs, while MFE¯and standard deviation ( σ) models are invoked separately for z-score calculation (Eq. (1)). All training code was run through Google Colab (Bisong, 2019) and can be viewed and run directly in the corresponding python notebook (File S2) or using our fully functional google colab page (https://colab.research.google.com/github/moss-lab/ScanFold2.0/blob/main/SF2_notebook.ipynb).

(1) z−score=MFE−MFE¯σ

Updates to ScanFold 2.0 and integration in the webserver

To make the use of SF2 more user-friendly, it has been incorporated into the Moss Lab Tools webserver (https://mosslabtools.bb.iastate.edu/scanfold2). Similar to SF1, any sequence longer than the chosen window size can be uploaded (or pasted) in FASTA format, all parameters can be set by the user, and the scan can be started by clicking the submit button at the bottom of the page. Once the prediction is complete the results are output in an Integrative Genomics Viewer (IGV.js) window (Robinson et al., 2020) and made available for download as a zip file.

Testing of ScanFold 2.0 vs ScanFold 1.0

SF2 was tested to determine its accuracy and speed compared to that of SF1. Testing was performed on HIV-1, Zika, and SARS-CoV-2 genomes, which had been previously analyzed using SF1 (Andrews, Baber & Moss, 2020; Andrews et al., 2021; Andrews, Roche & Moss, 2018). To ensure that our testing was comprehensive we compared SF2 mono- and dinucleotide shuffling results to those of SF1 mono- and dinucleotide shuffling using 100, 1,000, and 10,000 randomizations for each genome. The results of all output CT files (i.e., −2, −1, and No Filter z-scores) from both versions of ScanFold were compared using an in-house python script, ct_compare.py (Andrews et al., 2021; https://github.com/moss-lab/SARS-CoV-2). This comparison allowed us to evaluate the percent of paired nucleotides and the percent similarity or consistency between the output files of both versions of ScanFold as well as determine the improvements in speed for each run. Additionally, we were able to compare the outputs from SF1 (mono- vs dinucleotide shuffling and different number of randomizations) and the outputs of SF2 (mono- vs dinucleotide shuffling) to themselves to evaluate their performance using different shuffling methods. In total, 13 different comparisons were completed for each genome. All accuracy and speed results can be found in File S1.

ROC curve analysis

Receiver Operating Characteristic (ROC) curve analysis was performed on ScanFold-Fold results for SF1 mono- and dinucleotide shuffling using 100 and 10,000 randomizations as well as SF2 mono- and dinucleotide shuffling models following a previously establish protocol (Andrews et al., 2021). Briefly, reactivity value thresholds were sequentially set from the lowest to highest value at 1% intervals (i.e., 0–100% constrained) for various SHAPE and DMS reactivity datasets generated from SARS-CoV-2 probing experiments (Huston et al., 2021; Lan et al., 2021; Manfredonia et al., 2020; Sun et al., 2021). The −1 z-score CT files from SF1 and SF2 were cross referenced to the constrained reactivity threshold datasets and used to find the true positive rate (TPR) and false positive rates (FPR) for each comparison. In this analysis, the TPR and FPR are represented by Eqs. (2) and (3) below:

(2) TPR=TP(TP+FN)

(3) FPR=FP(FP+TN)

The true positive (TP) is defined as being paired in the ScanFold −1 z-score CT file and paired at the defined reactivity threshold. The false negative (FN) is defined as being paired in the ScanFold −1 z-score CT file and unpaired at the reactivity threshold. The false positive (FP) is defined as being unpaired in the ScanFold −1 z-score CT file and paired at the reactivity threshold. The true negative (TN) is defined as being unpaired in the ScanFold −1 z-score CT file and unpaired at the given reactivity threshold. When the reactivity threshold is set to 0%, the TPR and FPR will equal zero, and when the reactivity threshold is set to 100%, the TPR and FPR will equal one. Thus, if a ScanFold predicted RNA secondary structure model is truly random, when compared to increasing reactivity thresholds from a probing data set, then the TPR and FPR will increase proportionately and produce a linear trend in the plot and a small area under the curve (AUC). However, if the ScanFold predicted RNA secondary structure model agrees with the reactivity data set, the TPR will initially rise faster than the FPR, producing a curve on the plot and therefore a larger AUC. This allows for a quantitative assessment and comparison of each ScanFold predicted model’s ability to fit the data via their respective AUCs. All the ROC and AUC analyses can be found in File S3.

Results and discussion

Comparing time and accuracy of ScanFold 2.0 vs ScanFold 1.0

SF2 requires significantly less time than SF1 using only 100 randomizations, with increases in speed being even greater when compared to SF1 using 1,000 and 10,000 explicitly shuffled RNA sequences for z-score calculations. In both cases, increasing sequence length does increase the time needed, but this effect is seen to a lesser degree in SF2. When comparing the times, SF1 using 100 randomizations with mononucleotide shuffling takes 8.70, 1.02, and 1.75 h to complete all predictions for SARS-CoV-2, HIV, and Zika, respectively (Table 1). SF2 on the other hand reduces these times to 2.64, 0.27, and 0.35 h for SARS-CoV-2, HIV, and Zika, respectively (Table 2). This decrease in time for SF2 is greater for higher randomization numbers and dinucleotide shuffling (Tables 1 and 2). For SF2, the scanning step is now the fastest step in the process, taking only 0.27, 0.07, and 0.09 h for SARS-CoV-2, HIV, and Zika, respectively (Table 2). Importantly, increased speed does not come at the cost of reduced accuracy.

Table 1 Time required for SF1 runs using different shuffling methods and number of randomizations to finish.

The time required to finish runs for both versions of ScanFold were evaluated using different shuffling methods and number of randomizations. All times are reported in hours.

	Total time (h) 100 randomizations	Total time (h) 1,000 randomizations	Total time (h) 10,000 randomizations	
SARS SF1 Mono	8.70	21.28	164.17	
HIV SF1 Mono	1.02	4.58	32.85	
Zika SF1 Mono	1.75	4.15	36.55	
SARS SF1 Di	7.50	22.07	134.00	
HIV SF1 Di	0.95	4.48	35.58	
Zika SF1 Di	1.25	4.67	38.53	

Table 2 Time required for each step of SF2 to run, total SF2 run time, and increase in SF2 speeds compared to SF1.

The time required to finish SF2 scanning step, folding step, and both steps were evaluated using different shuffling methods. Increase in speed was calculated by dividing SF1 total run time for each shuffling technique at each number of randomizations by SF2 total run time. All times are reported in hours.

	Scan time (h)	Fold time (h)	Total time (h)	Speed increase 100 rand.	Speed increase 1,000 rand.	Speed increase 10,000 rand.	
SARS SF2 Mono	0.27	2.37	2.64	3.30×	8.06×	62.19×	
HIV SF2 Mono	0.07	0.20	0.27	3.78×	16.96×	121.67×	
Zika SF2 Mono	0.09	0.27	0.35	5.00×	11.86×	104.43×	
SARS SF2 Di	0.33	1.67	2.00	3.75×	11.04×	67.00×	
HIV SF2 Di	0.07	0.17	0.24	3.96×	18.67×	148.25×	
Zika SF2 Di	0.09	0.23	0.32	3.91×	14.59×	120.41×	

Gross comparisons of the percent of predicted pairs by SF1 and SF2 using 100, 1,000, and 10,000 randomizations with mononucleotide shuffling displays an average difference of 2.00% (0.03% to 4.5%) between all z-score cutoffs across the three genomes analyzed, regardless of the number of randomizations. HIV is the most consistent between versions, displaying less than a 1.25% difference in −2 z-score pairs, 3.2% difference in −1 z-score pairs, and 0.5% difference in all pairs (no filter) across all randomizations (File S1). In a similar analysis, it is also seen that the percent similarity or consistency of paired and unpaired nucleotides between SF1 and SF2 using mononucleotide shuffling is quite high, with the average difference being only 4.01% (1.11% to 6.29%) between all z-score cutoffs across the three genomes analyzed (File S1). Here, HIV shows some of the best results with only the no filter cutoff reaching a 6.24% difference, and z-score cutoffs of −2 and −1 being only 1.42% and 4.7% different, respectively (Fig. 2).

Figure 2 SF1 and SF2 comparisons of HIV results.

Comparison of SF1 and SF2 percent similarity in paired and unpaired nucleotides using mono and dinucleotide shuffling with 100, 1,000, and 10,000 randomizations. (A) HIV percent similarity in −2 z-score results. (B) HIV percent similarity in −1 z-score results. All comparison were done using SF1 results as the reference and SF2 results as the target for comparison.

The same analyses were carried out between SF1 and SF2 using dinucleotide shuffling. Comparing the percentage of predicted paired nucleotides using 100, 1,000, and 10,000 randomizations with dinucleotide shuffling displays an average difference of 5.26% (0.57% to 10.26%) between all z-score cutoffs across the three genomes analyzed. HIV showed the least variance with a 4.38% difference in −2 z-score pairs, an 8.72% difference in −1 z-score pairs, and a 1.85% difference in all (no filter) pairs across all randomizations (File S1). The percent similarity or consistency in the paired and unpaired nucleotides between SF1 and SF2 using dinucleotide shuffling is again quite high, especially for structures within the significant z-score cutoffs of −2 and −1, with the average difference being 10.42% (4.71% to 20.64%) between all z-score cutoffs across the three genomes analyzed (File S1). Here, HIV shows some of the best results with only the no filter cutoff reaching a 20.64% difference, and z-score cutoffs of −2 and −1 being only 4.82% and 10.16% different, respectively (Fig. 2). Notably, when comparing the predictions to biochemical probing data, all approaches showed consistency with experimental results (Fig. 3).

Figure 3 ROC analysis of SF1 and SF2 results.

ROC analysis of six different in vivo and in vitro SHAPE and DMS biochemical probing dataset of the SARS-CoV-2 genome. (A) Plot of the initial ROC analysis curve with the AUC for SF1 using mono and dinucleotide shuffling at 100 and 10,000 randomization and SF2 results using mono and dinucleotide shuffling for Lan et al. (2021) DMS in vivo dataset. SF1 mononucleotide with 100 randomizations in blue (AUC = 0.776), SF1 mononucleotide with 10,000 randomizations in orange (AUC = 0.773), SF1 dinucleotide with 100 randomizations in gray (AUC = 0.759), SF1 dinucleotide with 10,000 randomizations in yellow (AUC = 0.758), SF2 mononucleotide in black (AUC = 0.780), and SF1 dinucleotide in green (AUC = 0.773). (B) Plot of the ROC analysis with the AUC for SF1 using mono and dinucleotide shuffling at 100 randomizations and SF2 results using mono and dinucleotide shuffling for all probing datasets. All SF1 and SF2 results for Lan et al. (2021) DMS in vivo in blue (AUC = 0.759–0.780), Manfredonia et al. (2020) DMS in vitro in yellow (AUC = 0.722–0.741), Sun et al. (2021) SHAPE in vivo in green (AUC = 0.725–0.748), Manfredonia et al. (2020) SHAPE in vitro in orange (AUC = 0.677–0.691), Manfredonia et al. (2020) SHAPE in vivo in gray (AUC = 0.660–0.678), and Huston et al. (2021) SHAPE in vivo in black (AUC = 0.622–0.633).

Mono vs Di nucleotide shuffling of ScanFold 2.0

When comparing SF1 and SF2 results for mononucleotide shuffling there is an average difference in percent paired of 2.00% (0.03% to 4.5%) and in the majority of cases SF2 is predicting more pairs than SF1. For all results other than HIV and SARS-CoV-2 all pairs (no filter), SF2 consistently predicts more pairs than SF1. When comparing SF1 and SF2 results for dinucleotide shuffling, there is an average difference of 5.26% (0.57% to 10.26%) and similar to mononucleotide shuffling, all results other than Zika no filter (all pairs), show that SF2 is always predicting slightly more pairs. These small differences serve as evidence that SF1 and SF2 are producing an almost identical number of pairs when the same shuffling method is used (File S1).

When comparing the results of SF1 mononucleotide shuffling to SF1 dinucleotide shuffling, on average, mononucleotide shuffling finds more pairs than dinucleotide shuffling, but this does not always hold true—as is the case with all iterations of Zika results for all pairs (no filter; Fig. S1). For SF1, the smallest difference between results is seen in Zika (all pairs), where dinucleotide shuffling finds 0.72% more pairs than mononucleotide, while the largest difference is seen in Zika −1 z-score pairs, where mononucleotide shuffling predicts 8.65% more pairs than dinucleotide (Table S1 and Fig. S1). SF2 comparisons show a split between which shuffling method predicts more pairs. In the case of Zika, the same trend seen for SF1 holds true for SF2, with mononucleotide shuffling finding more pairs than dinucleotide shuffling for all cutoffs other than all pairs. For HIV, SF2 dinucleotide shuffling finds more pairs than mononucleotide shuffling at all z-score cutoffs, but for SARS-CoV-2, dinucleotide shuffling finds more pairs than mononucleotide shuffling only at the −2 z-score cutoff. Here, the smallest difference in SF2 is seen in the SARS-CoV-2 results for all pairs where mononucleotide shuffling finds 0.36% more pairs than dinucleotide, and the largest difference is seen in Zika results for −1 z-score pairs where mononucleotide shuffling finds 3.13% more pairs than dinucleotide (Table S1 and Fig. S1). These small variations between the shuffling methods provide further evidence that SF1 and SF2 are performing similarly in identifying ordered structure, and that the shuffling technique used does not influence the results to a high degree.

As additional evidence that the shuffling method does not have a large impact on results, we analyzed the percent consistency in pairing between SF1 and SF2 using 100, 1,000, and 10,000 randomizations with both shuffling methods. Here, we observe that SF2 mono- and dinucleotide results are generally consistent (within 5–10%) with that of SF1 mono- and dinucleotide results across all three genomes, with HIV demonstrating the most consistency (Fig. 2). The general trend among the three genomes shows the more stringent −2 and −1 z-score predictions are always within 10–12% consistency of each other regardless of shuffling or randomization, while the no filter pairings often show more variation (File S1). All comparisons indicate little benefit of using dinucleotide over mononucleotide shuffling, as the percent consistency between these methods in both SF1 and SF2 predictions are on average 7.53% different (1.85% to 18.27%) and, when looking at just SF2 using both methods, predictions are on average 4.79% different (1.96% to 9%) (File S1). The differences associated with SF1 and SF2 mononucleotide and dinucleotide shuffling can most likely be equated to the differences in z-scores (Fig. S2 and File S4). The box and whisker plot in Fig. S2 show that for SF2, the average z-scores are consistently lower for both shuffling methods compared to that of SF1, and the differences in z-scores between the two shuffling methods is also much smaller for SF2 (average difference of −0.019) compared to that of SF1 (average difference of −0.363) (Table S2 and File S4). The lower overall z-score of SF2 is potentially causing the differences in percent paired and percent similarity or consistency that is seen between the shuffling methods when comparing SF1 and SF2. Regardless of the differences in percent paired, percent similarity or consistency, and z-score, the results of mononucleotide and dinucleotide shuffling for SF2 are similar to SF1 as shown by the agreement of biochemical probing data (Fig. 3).

ROC analysis of SARS-CoV-2

As another layer of validation, we followed an established protocol (Andrews et al., 2021) to perform a receiver operator characteristic (ROC) analysis on the SARS-CoV-2 genome predictions. We compared SF1 and SF2 results using 100 to 10,000 randomizations with both shuffling methods, to six different SHAPE and DMS biochemical probing datasets (Huston et al., 2021; Lan et al., 2021; Manfredonia et al., 2020; Sun et al., 2021). Here, the effect of increasing the stringency of reactivity cutoffs, which considers whether a site is to be paired in the model, provides a measure of the consistency of probing data compared to ScanFold models (see Material and Methods and (Andrews et al., 2021)). We initially compared the SF1 results using both shuffling methods with 100 and 10,000 randomizations and the SF2 results using both shuffling methods to the Lan et al. (2021) in vitro DMS data. The ROC analysis showed that all SF1 and SF2 results clustered into the same curve with almost identical area under the curve (AUC) values (Fig. 3A). The ROC analysis of SF1 and SF2 results using 100 randomizations and both shuffling methods was repeated on all six probing datasets. SF2 predictions match the curves of both the previous analysis and all SF1 results obtained in this study (Fig. 3B). After calculating the area under the AUC for each set of results, all were found to be above 0.5, indicating global consistency of the data with SF1 and SF2 results. AUC values for SF2 ranged from a minimum value of 0.629 for comparison of SF2 dinucleotide to in vivo SHAPE dataset (Huston et al., 2021) to a maximum value of 0.780 for comparison of SF2 mononucleotide to in vivo DMS dataset (Lan et al., 2021). No large differences were observed when comparing any of the AUC values between SF1 or SF2 and the respective datasets. These findings indicate that, similar to SF1, SF2 is detecting the most robust local elements that do not vary between experimental conditions.

Selection of ScanFold parameters

As with all scanning window analyses, it is important to understand the impact of selecting different window and step sizes. In general, the use of 120 nt windows has been found to be optimal as it allows for the identification of local structures and reduces the prediction of spurious longer-range interactions (Lange et al., 2012). In previous studies, we have found that window sizes from 60–150 nt yield the results most consistent with structures informed by probing data (Andrews, Baber & Moss, 2020; Andrews et al., 2021; Andrews, Roche & Moss, 2018). By decreasing the window size, the predicted structures will be smaller overall, as no structure larger than the window will be found, which can aid in predicting smaller local structures. For detection of larger structures, using a window size up to 600 nt has been shown to successfully identify larger structures in the HIV genome (Andrews et al., 2021; Andrews, Roche & Moss, 2018), however, we would recommend this as a separate analysis; larger window sizes (>200 nt) tend to predict multiple structures per window, leading to a loss of fidelity when determining which specific structures yielded unusual stability metrics (see Supplemental of Andrews et al., 2021). Therefore, a 120 nt window is ideal, unless one is specifically interested in smaller or larger structures. The other important consideration is step size. Using a single nucleotide step size provides the highest resolution scans and the most robust data, as each nucleotide is refolded across all potential overlapping windows. Larger steps can “cut into” structured regions leading to less windows for analysis and more ambiguous results. With the added speed of SF2, a 1-nt step size is now possible for longer sequences. For most applications, a 120 nt window, 1-nt step size, and mononucleotide shuffling are recommended. When interpreting the ScanFold results, the z-score cutoff is an important value to consider. The use of a strict −2 z-score cutoff provides structures with the most potential to be ordered for stable folding, whereas a −1 cutoff yields more, less significantly ordered, structures for consideration. Regardless, ScanFold predictions should be considered initial structural hypotheses ahead of additional work (e.g., comparative analyses, structure probing, and functional assays).

Conclusion

SF2 produces effectively indistinguishable results to that of SF1 in a fraction of the time. The implementation of a machine learning approach has also eliminated the need to optimize the number of randomizations for each scan. Based on our results, we see that SF2 using the dinucleotide shuffling model tends to produce results more similar to mononucleotide than SF1; however, both SF1 and SF2 results are generally similar to each other, with slight bias toward lower z-scores in SF2 arising from the very large training data sets used (likely reflecting more accurate z-score values). ROC analysis using several SHAPE and DMS datasets against SF1 and SF2 predictions also suggests that, regardless of the model, SF2 detects robust structural elements that persist between experimental conditions. Here, we have demonstrated that the improved SF2 algorithm performs similarly to SF1 but in a fraction of the time. Although this analysis was focused on viral genomes, SF2 has general applicability for any RNA sequence of interest. We hope that this improved speed can provide the RNA community with a fast, accurate, and user-friendly tool that will help in finding potentially functional structures across any gene or transcript of interest and drive forward RNA research.

Supplemental Information

Supplemental Information 1 Comparison of SF1 and SF2 ct files.

The results from all CT files comparisons for HIV-1, ZIKA, and SARS-CoV-2. These comparisons include SF1 to SF2, SF1 to SF1, and SF2 to SF2 using both shuffling methods and 100, 1,000, and 10,000 randomizations. Additionally, the results are summarized for each genome in a separate tab of the file, which includes all the raw comparisons, tables, and calculated differences in percent of paired nucleotides for all conditions, and SF1 and SF2 run time tables and bar charts.

Click here for additional data file.

Supplemental Information 2 SF2 python notebook training code.

All training code for SF2 that was ran in Google Colab. This Python notebook can used to view and run all training code.

Click here for additional data file.

Supplemental Information 3 ROC analysis of all SF1 and SF2 results.

The results of the ROC analysis displayed in Figure 2. This includes all tpr, fpr, and AUC calculations for every dataset and all SF1 and 2 conditions used. Both of the ROC curve plots can also be found at the end of the document.

Click here for additional data file.

Supplemental Information 4 All SF1 and SF2 per window z-scores.

The scanning “.out” output data from all SF1 and SF2 run conditions for HIV-1, ZIKA, and SARS-CoV-2. All of the per window z-score data from each of the .out results were added to separate tabs genome tabs and average values were calculated. These values can be found in the last 4 tabs of the excel document.

Click here for additional data file.

Supplemental Information 5 Percent of nucleotides paired and percent similarity in pairing comparing SF1 to SF1 and SF2 to SF2.

The percent of nucleotides paired and the percent similarity in pairing from the comparison of SF1 to SF1 and SF2 to SF2 using both shuffling methods as well as 100, 1,000, and 10,000 randomizations for SF1. All plots on the left side are percent paired and all plots on the right are percent similarity in pairings. From top to bottom (A-C), the plots show results for SARS-CoV-2, ZIKA, and HIV-1. All plots are organized as follows from left to right: SF1 mono- to di- 100 randomizations for -2 z-score cutoff, SF1 mono- to di- 1000 randomizations for -2 z-score cutoff, SF1 mono- to di- 10000 randomizations for -2 z-score cutoff, SF2 mono- to di- for -2 z-score cutoff, SF1 mono- to di- 100 randomizations for -1 z-score cutoff, SF1 mono- to di- 1000 randomizations for -1 z-score cutoff, SF1 mono- to di- 10000 randomizations for -1 z-score cutoff, SF1 mono- to di- for -1 z-score cutoff, SF1 mono- to di- 100 randomizations for no filter z-score cutoff, SF1 mono- to di- 1000 randomizations for no filter z-score cutoff, SF1 mono- to di- 10000 randomizations for no filter z-score cutoff, and SF2 mono- to di- for no filter z-score cutoff.

Click here for additional data file.

Supplemental Information 6 Average SF1 and SF2 per window z-scores for all analyzed genomes.

Box and whisker plot of the average window z-score for all genomes analyzed. The plots show that in general mononucleotide shuffling produces lower z-scores, dinucleotide shuffling produces higher z-scores, and SF2 mono- and dinucleotide shuffling produce more similar z-scores than SF1 mono- and dinucleotide shuffling. From top to bottom (A-C) the plots represent ZIKA, HIV, and SARS-CoV-2. Each plot is organized in the same way. From left to right SF1 100 randomizations with mono-, SF1 100 randomizations with di-, SF1 1000 randomizations with mono-, SF1 1000 randomizations with di-, SF1 10000 randomizations with mono-, SF1 10000 randomizations with di-, SF2 with mono-, and SF2 with di-.

Click here for additional data file.

Supplemental Information 7 Percent difference in percent of predicted paried nucleotides comparing SF1 to SF1 mono and dinucleotide shuffling and SF2 to SF2 mono and dinucleotide shuffling.

Each cell in the table is the percent difference in percent of predicted nucleotides that are paired. Positive values indicate mono- predicted more pairs and negative values indicate that di- predicted more pairs for the respective genome, version of SF, randomizations, and shuffling method. From left to right SF1 100 randomization difference in percent paired for -2 z-score pairs, SF1 1000 randomization difference in percent paired for -2 z-score pairs, SF1 10000 randomization difference in percent paired for -2 z-score pairs, SF2 difference in percent paired for -2 z-score pairs, SF1 100 randomization difference in percent paired for -1 z-score pairs, SF1 1000 randomization difference in percent paired for -1 z-score pairs, SF1 10000 randomization difference in percent paired for -1 z-score pairs, SF2 difference in percent paired for -1 z-score pairs, SF1 100 randomization difference in percent paired for no filter z-score pairs,SF1 1000 randomization difference in percent paired for no filter z-score pairs, SF1 10000 randomization difference in percent paired for no filter z-score pairs, and SF2 difference in percent paired for no filter z-score pairs. From top to bottom ZIKA, HIV-1, SARS-CoV-2 genomes that coincide with the differences in percent paired nucleotides.

Click here for additional data file.

Supplemental Information 8 Average z-scores from all ScanFold analysis windows for SF1 and SF2 using both shuffling methods and different randomizations.

The average z-scores for all windows in every ScanFold run were determined and the average difference between shuffling techniques was calculated from all analyzed genomes. From left to right SF1 using 100 randomization mono-, SF1 using 100 randomizations di-, difference between SF1 using 100 randomizations with mono- and dinucleotide, SF1 using 1000 randomizations with mono-, SF1 using 1000 randomizations with di-, difference between SF1 using 1000 randomizations with mono- and dinucleotide, SF1 using 10000 randomizations with mono, SF1 using 10000 randomizations with di-, difference between SF1 using 10000 randomizations with mono- and dinucleotide, SF2 mono-, SF2 di-, and difference between SF2 with mono- and dinucleotide. From top to bottom ZIKA, HIV, SARS-CoV-2, and the average difference in z-score between shuffling techniques.

Click here for additional data file.

Thank you to the Iowa State University Research IT group for their support over the course of this project and members of the Moss Lab for their input.

Additional Information and Declarations

Competing Interests

Author Contributions

Data Availability

The authors declare that they have no competing interests.

Ryan J. Andrews conceived and designed the experiments, performed the experiments, analyzed the data, prepared figures and/or tables, authored or reviewed drafts of the article, and approved the final draft.

Warren B. Rouse conceived and designed the experiments, performed the experiments, analyzed the data, prepared figures and/or tables, authored or reviewed drafts of the article, and approved the final draft.

Collin A. O’Leary analyzed the data, prepared figures and/or tables, and approved the final draft.

Nicholas J. Booher performed the experiments, prepared figures and/or tables, and approved the final draft.

Walter N. Moss conceived and designed the experiments, analyzed the data, prepared figures and/or tables, authored or reviewed drafts of the article, and approved the final draft.

The following information was supplied regarding data availability:

The data and code are available in the Supplemental Files.

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
