# Peer review of "ScanFold 2.0: a rapid approach for identifying potential structured RNA targets in genomes and transcriptomes"

_PeerJ, doi:10.7717/peerj.14361_

## Round 0.1 · original submission · Minor Revisions

I would like to invite the authors to improve the content of the manuscript and to respond the reviewers' comments.

·

Basic reporting

No comment

Experimental design

No comment

Validity of the findings

no comment

Additional comments

The manuscript is scientifically sound and well written. Besides, it also provides a good tool for a rapid identification of structural and /or functional RNA elements in genomes and transcriptomes. SF2 algorithm requires less time than SF1 and it speeds is greater when compared to SF1. Additionally, SF2 predicts more pairs than SF1. Thus, SF2 produces effectively indistinguishable results than SF1 in a fraction of the time.
The following minor corrections should be considered.
Line 74, MFE should be in full to allow reader know what it stands for, then the MFE in parenthesis. Like what you did in line 104 with Rectified Linear Unit (ReLu)
Line 133 ROC should be in full and the abbreviation in parenthesis (ROC) or make it full in line 132.
NB: In a manuscript the first abbreviation must be in full. This will allow readers known what it means.
Lines 279, 283, Bertamini et al 2002a and 2005b. The letters must be deleted. It is only used when the first author publishes articles in the same year. Check also with Cosgrove et al., 2005a and 2015b. However, use it Only if it is the journal requirement

Reviewer 2 ·

Basic reporting

General impressions: The authors in ScanFold 2.0: A rapid approach for identifying potential structured RNA targets in genomes and transcriptomes presents an update to the ScanFold 1.0 algorithm. The algorithm computes a thermodynamic z-score based on minimum free energy (MFE) predictions of RNA folding compared to random sequences. ScanFold 2.0 improves upon ScanFold 1.0 by decreasing the computational time through a machine learning approach to calculate the randomized z-scores, while also maintaining similar accuracies of prediction to ScanFold 1.0. This allows ScanFold 2.0 to be more amenable to genomic/transcriptomic analyses compared to ScanFold 1.0, which is particularly beneficial for the study of long viral genomes, three of which the authors highlight in their manuscript. These improvements increase the utility of ScanFold 2.0, and I believe it will continue to be a valuable resource for the RNA community as its predecessor was. ScanFold 2.0 can also work in complement with other computational and experimental tools that analyze RNA structure. I therefore recommend the manuscript to be accepted with the following minor revisions.

1. The authors should fix the following grammar errors or wording to improve clarity before acceptance: add a comma before “as” (line 44), add a comma after “functional” (line 63), hyphen instead of slash between “structure/function” (line 52), consider splitting into shorter sentences for clarity (lines 55-63), choose either “impact” or “significance” (line 58), choose either “functional” or “evolved” (line 65), consider rephrasing these concepts to improve clarity (lines 69-72), define MFE as “minimum free energy (MFE) of folding” (line 67-68), remove the hashtag after each equation (lines 108, 142, 143), add hyphen in “user friendly” (line 110), refer to ScanFold as either ScanFold or ScanFold 1.0 to be consistent (line 116), add a comma before “proportionately” (line 152), add comma after “Zika” (lines 166, 167, 170), refer to SARS as “SARS-CoV-2” or define it (for example, lines 166, 167, 170), add a comma after “different” (line 194), add a comma after “data” (line 195), add a comma after “average” (line 207), rephrase this sentence to improve clarity and fix the grammatical error (lines 209-211), add a comma after “SARS” (line 216), add a comma before “and” (line 233), add a comma after “z-score” (line 243), consider rephrasing line 275 to “improved SF2 algorithm performs similarly to SF1 but in a fraction of the time,” and fix the spelling error of “paired” in Table S1.

2. The GitHub link to ScanFold 2.0 is broken (line 86). I assume the repository has not been made public yet.

3. In Figure 2, it would be helpful to have a right-hand axis in the plots to show how close the bars are to 100%.

Experimental design

1. The authors feature three examples of viral genomes (SARS-CoV-2, HIV, and Zika). Is ScanFold most suitable for long viral RNA genomes/transcriptomes? How does ScanFold perform for lncRNAs or mRNAs in SF1 vs. SF2? I recommend the authors describe whether ScanFold is of specific utility for genomes/transcriptomes (viral or otherwise) or is of general applicability to all RNAs.

Validity of the findings

1. Many of the results focus on comparisons between mononucleotide and dinucleotide shuffling. Could the authors briefly explain in the introduction the rationale behind these two shuffling methods and why they’re important?

2. In lines 194-196, could the authors comment more about the comparison to experimental results and the degree of consistency or lack thereof?

3. In line 204, why do the authors think SF2 predicts more pairs, and can any actionable conclusions be drawn from these additional predictions?

Additional comments

1. It would be helpful to ground the results in why choosing different parameters (shuffling method, filtering cutoff, number of randomizations) is significant and how a user would go about choosing the appropriate parameters for their RNA of interest. Can the author provide recommended guidelines for choosing parameters?

2. The Python notebook with the TensorFlow code is a fantastic, hands-on teaching resource. However, there seems to be a ValueError in the “Making predictions” section. The authors should check to make sure this code can run without errors on any computer.

---

## Round 0.2 · accepted · Accept

The authors have satisfactorily addressed the reviewers' comments. The authors however must correct some minor grammatical errors at the proof stage. I congratulate the authors for their work once again.

·

Basic reporting

No comment

Experimental design

No comment

Validity of the findings

Ni comment

Additional comments

The authors have addressed all my comments on the revised version

Reviewer 2 ·

Basic reporting

I recommend the authors read through their manuscript a few more times to catch final grammar errors. Otherwise, the authors have addressed all comments satisfactorily.

Experimental design

The authors have addressed all comments satisfactorily.

Validity of the findings

The authors have addressed all comments satisfactorily.

Additional comments

The paragraph "Selection of ScanFold parameters" is a very helpful addition. No further comments.